The BAF53A-BACH1-GCLM axis regulates glutathione metabolism and enhances ferroptosis resistance in esophageal squamous cell carcinoma

Jiang Weijuan
Zhang Jie
Chen Canjuan
Shi Jiangwei
Fan Lihua jjfanlihua@163.com
Department of Radiotherapy, Jingjiang People’s Hospital Affiliated to Yangzhou University , Taizhou, Jiangsu , China
Shi Huashan
Electronic publication date: 2025 Oct 3
Publication date: 2025
Volume: 13
Electronic Location ID: e20156
Received 2025 Jul 16; Accepted 2025 Sep 8
Copyright: © 2025 Jiang et al.
Copyright year: 2025
Copyright holder: Jiang et al.
License: This is an open access article distributed under the terms of the Creative Commons Attribution License, which permits unrestricted use, distribution, reproduction and adaptation in any medium and for any purpose provided that it is properly attributed. For attribution, the original author(s), title, publication source (PeerJ) and either DOI or URL of the article must be cited.
License URL: https://creativecommons.org/licenses/by/4.0/

Keywords: BAF53A, Esophageal squamous cell carcinoma, Ferroptosis, Glutathione metabolism

Funding: The authors received no funding for this work.

==============================
Objective

Esophageal squamous cell carcinoma (ESCC), a highly lethal malignancy, exhibits poor survival rates and limited treatment options. Ferroptosis, a regulated form of cell death driven by lipid peroxidation, emerges as a potential therapeutic target. However, the mechanisms suppressing ferroptosis in ESCC remain poorly understood.

Methods

Short hairpin RNA (shRNA) was employed to knock down BAF53A and BACH1 in ESCC cell lines, followed by assessments of cell proliferation, colony formation, and ferroptosis sensitivity. Glutathione (GSH) metabolism was evaluated by measuring GSH/GSSG and NADP+/NADPH ratios, reactive oxygen species (ROS) levels, and lipid peroxidation through flow cytometry and fluorescence imaging. Molecular interactions were evaluated using co-immunoprecipitation and chromatin immunoprecipitation sequencing (ChIP-seq) to identify transcriptional targets of the BAF53A-BACH1 complex.

Results

BAF53A was elevated in ESCC, and its depletion impaired cell proliferation and colony formation ability of cells. Knockdown of BAF53A disrupted GSH metabolism, leading to increased ROS levels, reduced GSH/GSSG and NADP+/NADPH ratios, and enhanced ferroptosis sensitivity. Mechanistically, BAF53A collaborated with BACH1 to transcriptionally activate glutamate-cysteine ligase modifier subunit (GCLM), a key enzyme in GSH biosynthesis. Overexpression of GCLM restored redox balance and cell viability in BAF53A- or BACH1-silenced cells.

Conclusions

The BAF53A-BACH1-GCLM axis constitutes a novel egulatory pathway that integrates chromatin remodeling, transcriptional regulatione, and ferroptosis resistance in ESCC. Targeting this axis may offer a promising approach to exploit metabolic vulnerabilities and enhance ferroptosis sensitivity in ESCC treatment.

Introduction

Esophageal squamous cell carcinoma (ESCC) is a highly lethal malignancy characterized by poor prognosis and limited therapeutic options (An, Li & Jia, 2023). Despite advancements in surgery, chemotherapy, and immunotherapy, the 5-year survival rate remains below 20%, highlighting the need for innovative treatment strategies (Sheikh et al., 2023). Tumor growth and progression in ESCC are driven by dysregulated transcriptional networks, epigenetic modifications, and metabolic adaptations that enable cancer cells to thrive under stress (Puhr, Prager & Ilhan-Mutlu, 2023). Among these adaptations, the regulation of oxidative stress and ferroptosis resistance has emerged as a key vulnerability in cancer cells (Deboever et al., 2024; Lander, Lander & Gibson, 2023).

Glutathione (GSH), the primary intracellular antioxidant, plays a central role in maintaining redox homeostasis and protecting cells from reactive oxygen species (ROS)-induced damage (Bansal & Simon, 2018; Niu et al., 2021). GSH metabolism is essential for detoxifying lipid peroxides, thereby preventing ferroptosis—a form of regulated cell death caused by lipid peroxidation and iron accumulation (Koppula, Zhuang & Gan, 2021). Dysregulation of GSH metabolism is frequently implicated in cancer progression (Asantewaa & Harris, 2021). For instance, increased GSH biosynthesis is linked to chemoresistance in ovarian cancer, increased tumor proliferation in breast cancer, and ferroptosis suppression in pancreatic cancer (Kennedy et al., 2020). Key enzymes regulating GSH metabolism, like glutamate-cysteine ligase catalytic subunit (GCLC) and glutamate-cysteine ligase modifier subunit (GCLM), are often upregulated in cancers, contributing to redox balance and tumor survival (Kalinina & Gavriliuk, 2020). Elucidating the molecular mechanisms governing GSH metabolism in ESCC may uncover novel therapeutic targets.

BRG1-associated factor 53A (BAF53A), also known as ACTL6A (actin like 6A), is a subunit of the SWI/SNF chromatin remodeling complex that regulates gene expression by altering chromatin structure (Vaicekauskaitė et al., 2022). Recognized as an oncogenic driver in cancers such as glioblastoma (Hu et al., 2022; Ji et al., 2018), hepatocellular carcinoma (Liu et al., 2023; Wang et al., 2022a), and lung cancer (Ma & Shan, 2021; Xiao, Lin & Lin, 2021), BAF53A promotes cell proliferation, stem-like characteristics, and therapy resistance. Beyond chromatin remodeling, BAF53A interacts with transcription factors and histone modifiers to activate oncogenic transcriptional programs (Zhang et al., 2018). However, its role in regulating redox homeostasis and ferroptosis in ESCC remains uninvestigated.

BTB and CNC homology 1 (BACH1), a transcription factor crucial for oxidative stress regulation (Padilla & Lee, 2021), controls the expression of genes involved in antioxidant defense and GSH metabolism (Nishizawa, Yamanaka & Igarashi, 2023). In lung and breast cancers, BACH1 enhances tumor progression by increasing oxidative stress tolerance and ferroptosis resistance (Lee et al., 2019; Lignitto et al., 2019; Wiel et al., 2019). Notably, BACH1 regulates GSH-related genes such as GCLM. The BACH1-GSH axis is instrumental in maintaining redox balance, enabling cancer cells to survive under high oxidative stress (Irikura et al., 2023; Nishizawa et al., 2020; Nishizawa, Yamanaka & Igarashi, 2023). This suggests that BACH1 may collaborate with chromatin regulators like BAF53A to modulate redox homeostasis in ESCC.

Given the roles of BAF53A and BACH1 in tumor progression and stress resistance in other cancers, it is hypothesized that BAF53A promotes ESCC growth by regulating GSH metabolism and suppressing ferroptosis through transcriptional activation of GCLM in cooperation with BACH1. Ferroptosis serves as a tumor-suppressive mechanism, particularly in cancers characterized by high ROS levels (Jiang, Stockwell & Conrad, 2021; Zeng et al., 2023). Cancer cells frequently evade ferroptosis by upregulating antioxidant pathways, including GSH biosynthesis, underscoring the therapeutic potential of targeting ferroptosis resistance (Chen et al., 2021; Ursini & Maiorino, 2020).

This study demonstrates that silencing BAF53A disrupts GSH homeostasis, increases lipid peroxidation, and sensitizes ESCC cells to ferroptosis, thereby impairing tumor growth. Furthermore, BAF53A interacts with BACH1 to transcriptionally activate GCLM, establishing a functional link between chromatin remodeling, redox balance, and ferroptosis resistance. Our findings uncover a novel role for the BAF53A-BACH1 axis in ESCC progression and highlight its potential as a therapeutic target to exploit metabolic vulnerabilities in ESCC.

Materials and Methods

Cell culture

Human ESCC cell lines KYSE150 (3101HUMTCHu236) and KYSE450 (GDC0633) from the Cell Bank of the Chinese Academy of Sciences were authenticated via short tandem repeat (STR) profiling and cultured in RPMI-1640 medium (Gibco, Waltham, MA, USA), supplemented with 10% fetal bovine serum (FBS; Gibco, Waltham, MA, USA) and 1% penicillin-streptomycin (Gibco, Waltham, MA, USA) at 37 °C in a humidified incubator with 5% CO2. Cell lines were tested for mycoplasma contamination before experiments. Passages 3–10 were used for all experiments to minimize phenotypic variation.

Artificial gene interference in cells

Short hairpin RNAs (shRNAs) targeting BAF53A or BACH1 were cloned into the pLKO.1 lentiviral vector, with a scrambled shRNA as a negative control. Lentiviruses were generated by co-transfecting 293T cells with shRNA plasmids, psPAX2, and pMD2.G using Lipofectamine 3000 (Thermo Fisher, Waltham, MA, USA). Viral supernatants were harvested 48 h post-transfection, filtered through a 0.45 μm membrane, and utilized to transduce KYSE150 and KYSE450 cells in the presence of 8 μg/mL polybrene (Sigma). After 24 h, stable cells were selected with 2 μg/mL puromycin for 72 h. Knockdown efficiency was validated by quantitative polymerase chain reaction (qPCR) and western blot (WB) analyses.

qPCR analysis

Total RNA was isolated with TRIzol reagent (Invitrogen, Waltham, MA, USA). Complementary was synthesized from 1 μg of RNA with the PrimeScript RT reagent kit (Takara, Shiga, Japan). qPCR was performed using SYBR Green Master Mix (Thermo Fisher, Waltham, MA, USA) on a LightCycler 96 System (Roche, Basel, Switzerland). Relative mRNA level was gauged using the 2−ΔΔCt method, with GAPDH as an internal control. Each sample was analyzed in triplicate.

WB analysis

Total protein was extracted using radio-immunoprecipitation assay lysis buffer (Thermo Fisher, Waltham, MA, USA) plus protease and phosphatase inhibitors (Roche, Basel, Switzerland). Protein concentrations were quantified by the bicinchoninic acid kit (Thermo Fisher, Waltham, MA, USA). Equal amounts (30 μg) of protein were separated by sodium dodecyl sulfate-polyacrylamide gel electrophoresis and loaded onto polyvinylidene fluoride membranes (Millipore, Burlington, MA, USA). Membranes were blocked with 5% non-fat milk for 1 h and incubated overnight with primary antibodies against BAF53A, BACH1, GCLM, KI67, proliferating cell nuclear antigen (PCNA), and GAPDH (1:1000 dilution; Cell Signaling Technology (CST)) at 4 °C. After washing, membranes were probed with HRP-conjugated secondary antibodies (1:5000; CST) for 1 h at ambient temperature. Signals were detected using enhanced chemiluminescence reagent (Bio-Rad, Hercules, CA, USA) and imaged with a ChemiDoc system (Bio-Rad, Hercules, CA, USA).

Cell counting kit-8 assays

Cell viability was determined using the CCK-8 (Dojindo). KYSE150 and KYSE450 cells (2 × 103 cells/well) were seeded in 96-well plates and treated with shRNA, NAC (100 μM), H2O2 (50 μM), or scrambled controls. At 0, 12, 24, and 72 h, 10 μL of CCK-8 reagent was added to each well for 2-h incubation at 37 °C. Absorbance at 450 nm was read on a microplate reader (Thermo Fisher, Waltham, MA, USA).

Colony formation assays

Cells were cultured at 500 cells/well in six-well plates for 14 days. Colonies were fixed in 4% paraformaldehyde for 15 min and stained with 0.1% crystal violet for 20 min. Colonies containing no less than 50 cells were counted under an inverted microscope. Colony formation efficiency was manifested as the ratio of colonies formed to cells seeded.

Lipid peroxidation assays

C11-BODIPY 581/591 (Thermo Fisher, Waltham, MA, USA) was employed to examine lipid peroxidation. Cells were treated with 10 μM erastin or/and 1 μM Fer-1 for 24 h, followed by incubation with 2 μM C11-BODIPY for 30 min. Fluorescence was detected on a BD LSRFortessa flow cytometer. Data were analyzed with FlowJo software to quantify lipid peroxidation.

Chromatin immunoprecipitation sequencing and motif analysis

Chromatin immunoprecipitation sequencing (ChIP-seq) data from the GSE216350 dataset were analyzed to identify BAF53A binding peaks at the GCLM promoter. Peaks were visualized using the Integrative Genomics Viewer. Binding motifs were identified in the JASPAR database (http://jaspar.genereg.net/) and cross-referenced with published BACH1-binding motifs.

Co-immunoprecipitation

Nuclear lysates were prepared using the NE-PER Nuclear and Cytoplasmic Extraction Kit (Thermo Fisher, Waltham, MA, USA). Lysates (500 μg) were incubated overnight with 2 μg of anti-BAF53A or anti-BACH1 antibodies at 4 °C and then added with protein A/G magnetic beads (Thermo Fisher, Waltham, MA, USA). Immunoprecipitates were washed, eluted, and analyzed by WB. Input lysates served as loading controls.

ROS measurement

ROS levels were assessed using 2′, 7′-dichlorofluorescin diacetate (Sigma, Kanagawa, Japan). Cells were treated with 50 μM H2O2 for 24 h and incubated with 10 μM DCFH-DA at 37 °C in the dark for 30 min. Fluorescence intensity was measured on a microplate reader (excitation/emission: 485/530 nm) or observed under a fluorescence microscope.

Spearman correlation analysis

Expression data for BAF53A, BACH1, and GCLM were retrieved from TCGA-ESCA using UCSC Xena. Spearman correlation coefficients were calculated using GraphPad Prism v9, and significance was defined as P < 0.05.

Statistical analyses

SPSS 21.0 (IBM Corp., Armonk NY, USA) and GraphPad Prism 8.01 were employed for statistical analysis and data plotting. Shapiro-Wilk test checked the normal distribution of data. Measurement data are manifested as mean ± standard deviation. Independent sample t-tests were adopted for pairwise comparisons, while one-way or two-way analysis of variance (ANOVA) was employed for multi-group comparisons, followed by Tukey’ post-hoc examinations. Statistical significance was set at P < 0.05.

Results

BAF53A silencing impairs growth of ESCC cells

In both the GSE20347 dataset and TCGA-ESCA database, BAF53A was found to be significantly overexpressed in ESCC tissues compared to adjacent normal tissues (Figs. 1A, 1B). Consistently, qPCR analysis showed that the BAF53A expression was upregulated in KYSE150 and KYSE450 cells compared to normal HET1A cells (Fig. 1C) To investigate its functional role, we employed shRNAs to knock down BAF53A expression in KYSE150 and KYSE450 cells (Figs. 1D, 1E). Silencing of BAF53A resulted in a marked reduction in cell viability at 72 h and a significant impairment of long-term colony-forming capacity over 14 days (Figs. 1F, 1G).

Figure 1 Knockdown of BAF53A inhibits ESCC cell growth.

(A) BAF53A expression levels in tumor and adjacent normal tissues from 17 ESCC patients in the GSE20347 dataset. (B) Expression levels of BAF53A in tumor tissues from TCGA-ESCA dataset compared to normal tissues from the GTEx database. (C) qPCR analysis of BAF53A expression in KYSE150 and KYSE450 cells and normal HET1A cells. (D, E) qPCR and WB analysis of BAF53A levels following shRNA-mediated knockdown of BAF53A in KYSE150 and KYSE450 cells. (F) Cell viability changes over 0, 12, 24, and 72 h in KYSE150 and KYSE450 cells treated with BAF53A shRNA or scrambled shRNA, assessed by the CCK-8 assay. (G) Colony formation assay evaluating the number of colonies formed by KYSE150 and KYSE450 cells over 14 days post-BAF53A knockdown. Data are presented as dot-and-whisker plots, where each dot represents an individual biological replicate. Statistical analysis was performed using two-way ANOVA followed by Tukey’s multiple comparison test; P < 0.05 was considered statistically significant. * < 0.05, ** < 0.01, **** < 0.0001.

BAF53A facilitates ESCC growth by modulating GSH metabolism

Notably, GSH-metabolic genes in BAF53A-silenced ESCC cells, including GCLM, GCLC, GPX2, GPX4, SLC7A11, SLC1A5, and GLS, were significantly downregulated (Fig. 2A). These genes play pivotal roles in glutathione biosynthesis, antioxidant defense, and amino acid transport. As an essential antioxidant, GSH impedes oxidative damage by neutralizing ROS, thereby converting NADPH to NADP+ and supporting cellular homeostasis and proliferation (Fig. 2B). BAF53A knockdown considerably decreased the GSH/GSSG and NADP+/NADPH ratios in KYSE450 and KYSE150 cells (Figs. 2C, 2D) and increased intracellular ROS levels, irrespective of H2O2 treatment (Fig. 2E). Notably, treatment with the ROS scavenger N-acetylcysteine (NAC) restored cell viability in BAF53A-depleted cells, further implicating oxidative stress as a critical downstream effector (Figs. 2F, 2G).

Figure 2 BAF53A promotes ESCC cell growth by regulating GSH metabolism.

(A) qPCR analysis of GSH metabolism-related genes (GCLM, GCLC, GPX2, GSS, GLS, GPX4, SLC7A11, SLC1A5) in KYSE150 and KYSE450 cells treated with BAF53A shRNA or scrambled shRNA. (B) Schematic of the GSH/GSSG cycle and its impact on oxidative stress and NADPH/NADP+ ratio. (C, D) Ratios of GSH/GSSG (C) and NADP+/NADPH (D) in KYSE150 and KYSE450 cells following BAF53A knockdown. (E) Relative ROS levels measured by DCFH-DA fluorescence in KYSE150 and KYSE450 cells treated with or without 50 μM H2O2 for 24 h. (F, G) Relative cell viability (F) and colony formation (G) in KYSE150 and KYSE450 cells treated with or without 100 μM NAC post-BAF53A knockdown. Data are presented as dot-and-whisker plots, where each dot represents an individual biological replicate. Statistical analysis was performed using two-way ANOVA followed by Tukey’s multiple comparison test; P < 0.05 was considered statistically significant.

BAF53A inhibits lipid peroxidation to suppress ferroptosis in ESCC Cells

Given the interdependence of GSH metabolism and ferroptosis, we investigated whether BAF53A knockdown renders ESCC cells susceptible to ferroptosis. Remarkably, treatment with the ferroptosis inhibitor ferrostatin-1 (Fer-1) notably restored viability in BAF53A-silenced cells, whereas inhibitors of apoptosis (z-VAD-FMK), necroptosis (necrostatin-1), and autophagy had no discernible effects (Fig. 3A). BAF53A-depleted cells exhibited heightened sensitivity to ferroptosis inducers such as H2O2, erastin, and buthionine sulfoximine (BSO), while their response to chemotherapeutics like doxorubicin and 5-fluorouracil remained unchanged (Figs. 3B–3F). Lipid peroxidation, a hallmark of ferroptosis, was assessed using C11-BODIPY staining and found to be significantly elevated following BAF53A knockdown, especially after erastin treatment. This effect was mitigated by Fer-1 (Fig. 3G). Collectively, these results suggest that BAF53A suppresses ferroptosis by regulating lipid peroxidation and GSH metabolism.

Figure 3 BAF53A inhibits lipid peroxidation to suppress ferroptosis in ESCC cells.

(A) Cell viability of KYSE150 and KYSE450 cells treated with BAF53A shRNA or scrambled shRNA and cultured with or without 1 μM ferrostatin-1 (Fer-1), 2 μM necrostatin-1, 5 μM ZVAD-FMK, or 2 μM chloroquine for 24 h. (B-F) Viability of KYSE150 and KYSE450 cells treated with BAF53A shRNA or scrambled shRNA and cultured with varying concentrations of BSO (B), erastin (C), H2O2 (D), doxorubicin (E), and etoposide (F) for 24 h. (G) Lipid peroxidation levels, assessed by C11-BODIPY fluorescence, in KYSE150 and KYSE450 cells treated with erastin (10 μM), Fer-1 (1 μM), or both for 24 h. (H) IHC staining of the lipid peroxidation marker 4-HNE in xenograft tumors derived from KYSE150 and KYSE450 cells. Data are presented as dot-and-whisker plots, where each dot represents an individual biological replicate or mouse. Statistical analysis was performed using two-way ANOVA followed by Tukey’s multiple comparison test; P < 0.05 was considered statistically significant.

BAF53A regulates GCLM transcription via interaction with BACH1

The mRNA and protein levels of GCLM were found to be markedly downregulated in BAF53A-silenced cells (Figs. 4A, 4B). ChIP-seq analysis from the GSE216350 dataset revealed a prominent BAF53A-binding peak at the transcription start site (TSS) of GCLM (Fig. 4C). Motif analysis using JASPAR identified a sequence within this peak highly homologous to the BACH1-binding motif (Fig. 4D). Given that BACH1 is a known transcriptional regulator of GSH-related genes and a modulator of ferroptosis(Irikura et al., 2023; Nishizawa et al., 2020; Nishizawa, Yamanaka & Igarashi, 2023), we hypothesized that BAF53A interacts with BACH1 to regulate GCLM transcription. BACH1 knockdown visibly reduced GCLM expression (Figs. 4E, 4F); however, simultaneous knockdown of both BAF53A and BACH1 did not further suppress GCLM levels (Figs. 4G, 4H). Co-IP assays confirmed the physical interaction between BAF53A and BACH1, and nuclear colocalization was validated through immunofluorescence (Figs. 4I, 4J). To further confirm the binding of BACH1 or BAF53A to the GCLM promoter, ChIP analysis was conducted in BACH1-knockdown or BAF53A-knockdown cells to examine the interaction of BAF53A and BACH1 with the GCLM promoter. The results demonstrated that knockdown of BACH1 significantly reduced the binding of BAF53A to the GCLM promoter, and vice versa (Fig. 4K).

Figure 4 BAF53A regulates GCLM transcription via interaction with BACH1.

(A, B) qPCR and WB analysis of GCLM mRNA and protein levels in KYSE150 and KYSE450 cells following BAF53A knockdown. (C) ChIP-seq analysis showing BAF53A and IgG binding peaks near the GCLM promoter. (D) Predicted BACH1-binding motif identified within the BAF53A-binding peak at the GCLM promoter using JASPAR. (E, F) qPCR and WB analysis of GCLM and BACH1 expression following BACH1 knockdown in KYSE150 and KYSE450 cells. (G, H) qPCR analysis of GCLM expression in KYSE150 and KYSE450 cells with simultaneous knockdown of BAF53A and BACH1 (G) or simultaneous knockdown of BACH1 and BAF53A (H). (I) Co- IP of BAF53A and BACH1, showing reciprocal binding in KYSE150 and KYSE450 cells. (J) Fluorescence colocalization analysis of BAF53A and BACH1 in KYSE150 and KYSE450 cells. (K) Using anti -BAF53A or anti-BACH1 for ChIP, followed by qPCR analysis to examine the enrichment of the GCLM promoter. Data are presented as dot-and-whisker plots, where each dot represents an individual biological replicate. Statistical analysis was performed using two-way ANOVA followed by Tukey’s multiple comparison test; P < 0.05 was considered statistically significant.

BACH1 depletion restricts ESCC growth by disrupting GSH homeostasis

Knockdown of BACH1 in ESCC cells led to a significant and time-dependent reduction in cell viability. At 72 h, cell viability was decreased by over 50% in both KYSE450 and KYSE150 cells compared to controls (Fig. 5A). Clonogenic assays further confirmed a marked reduction in colony-forming ability, with colony counts reduced by approximately 60–70% following BACH1 silencing (Fig. 5B). Consistently, intracellular ROS levels, as assessed by DCFH-DA staining, were substantially increased in BACH1-depleted cells (Fig. 5C), accompanied by a 40–60% decrease in GSH/GSSG ratio (Fig. 5D) and over 50% decrease in NADP+/NADPH ratio (Fig. 5E). Lipid peroxidation levels, a hallmark of ferroptosis, were significantly elevated upon BACH1 knockdown (Fig. 5F). Moreover, BACH1-depleted cells exhibited increased sensitivity to ferroptosis inducers such as RSL3 and erastin, as reflected by a leftward shift in dose-response curves and lower IC50 values (Figs. 5G–5H). Sensitivity to H2O2-induced oxidative stress was also enhanced (Fig. 5I), collectively underscoring the crucial role of BACH1 in regulating redox homeostasis and promoting ESCC cell survival.

Figure 5 GCLM overexpression rescues growth defects in BAF53A- or BACH1-depleted ESCC cells.

(A, B) qPCR and WB analysis of GCLM expression in KYSE150 and KYSE450 cells overexpressing GCLM under BAF53A or BACH1 knockdown conditions. (C, D) Cell viability (C) and colony formation (D) in KYSE150 and KYSE450 cells overexpressing GCLM under BAF53A or BACH1 knockdown conditions. (E, F) Ratios of GSH/GSSG (E) and NADP+/NADPH (F) in KYSE150 and KYSE450 cells after GCLM overexpression. (G, H) ROS levels (G) and lipid peroxidation (H) in KYSE150 and KYSE450 cells after GCLM overexpression. (I) Spearman correlation analysis of GCLM expression with BAF53A and BACH1 in the TCGA-ESCA dataset. Data are presented as dot-and-whisker plots, where each dot represents an individual biological replicate. Statistical analysis was performed using two-way ANOVA followed by Tukey’s multiple comparison test; P < 0.05 was considered statistically significant.

GCLM overexpression rescues growth in BAF53A- or BACH1-silenced ESCC cells

To confirm the functional relevance of BAF53A-BACH1-mediated regulation of GCLM, we overexpressed GCLM in BAF53A- or BACH1-silenced cells (Figs. 6A, 6B). GCLM overexpression restored cell viability, enhanced colony formation capacity, and normalized GSH/GSSG and NADP+/NADPH ratios (Figs. 6C–6F). Furthermore, overexpression of GCLM mitigated ROS accumulation and lipid peroxidation (Figs. 6G, 6H). Correlation analysis of TCGA-ESCA data revealed a strong positive association between GCLM expression and levels of both BAF53A and BACH1 (Fig. 6I).

Figure 6 BACH1 knockdown restricts ESCC cell growth by disrupting GSH homeostasis.

(A, B) Cell viability (A) and colony formation (B) in KYSE150 and KYSE450 cells treated with BACH1 shRNA or scrambled shRNA. (C) ROS levels, measured by DCFH-DA fluorescence, in KYSE150 and KYSE450 cells following BACH1 knockdown. (D, E) Ratios of GSH/GSSG (D) and NADP+/NADPH (E) in KYSE150 and KYSE450 cells post-BACH1 knockdown. (F) Lipid peroxidation levels, assessed by C11-BODIPY fluorescence, in KYSE150 and KYSE450 cells following BACH1 knockdown. (G–I) Cell viability of KYSE150 and KYSE450 cells treated with BACH1 shRNA or scrambled shRNA and cultured with varying concentrations of BSO (G), erastin (H), and H2O2 (I). Data are presented as dot-and-whisker plots, where each dot represents an individual biological replicate. Statistical analysis was performed using two-way ANOVA followed by Tukey’s multiple comparison test; P < 0.05 was considered statistically significant.

Discussion

This study sought to unravel the molecular mechanisms driving ESCC progression, focusing on BAF53A’s role in GSH metabolism and ferroptosis regulation. Our findings demonstrate that BAF53A interacts with BACH1 to transcriptionally activate GCLM, a prominent enzyme in GSH biosynthesis. This BAF53A-BACH1-GCLM axis is critical in keeping redox homeostasis, suppressing lipid peroxidation, and protecting ESCC cells from ferroptosis, thereby driving tumor growth. These results provide views for the interplay between chromatin remodeling, transcriptional regulation, and metabolic reprogramming in ESCC. Our work also provides a comparative perspective on ferroptosis regulation in different cancers. Ferroptosis, a mode of regulated cell death resulting from iron-dependent lipid peroxidation, has been recognized as a tumor-suppressive mechanism. Previous studies have identified GPX4, SLC7A11 (xCT), and FSP1 as key regulators of ferroptosis in various cancers (Chen et al., 2021; Jiang, Glandorff & Sun, 2024; Ursini & Maiorino, 2020). While these studies primarily focused on post-transcriptional or enzymatic regulation of ferroptosis, our research highlights a transcriptional mechanism involving the BAF53A-BACH1-GCLM axis. This transcriptional control of GSH biosynthesis complements existing models of ferroptosis regulation and suggests new therapeutic opportunities for targeting ferroptosis resistance in ESCC.

BAF53A, also known as ACTL6A, is extensively studied in other cancers for its action in chromatin remodeling and transcriptional regulation 15770676. In glioblastoma, BAF53A drives tumor progression by maintaining stem cell-like properties through chromatin remodeling and activation of oncogenic transcriptional programs (Ji et al., 2018). In hepatocellular carcinoma, BAF53A has been shown to enhance proliferation and inhibit differentiation via interactions with the SWI/SNF complex (He et al., 2024; Xiao et al., 2016). Similarly, in lung cancer, BAF53A promotes epithelial-to-mesenchymal transition and metastatic potential (Sun et al., 2017). Our findings extend these observations by identifying a novel role for BAF53A in regulating ferroptosis. While previous studies focused on its contributions to proliferation and differentiation, our research highlights its role in metabolic adaptation through its interaction with BACH1, specifically targeting GSH metabolism to maintain redox balance and suppress ferroptosis.

BACH1 has emerged as a key transcriptional regulator of oxidative stress and redox homeostasis (Hayes, Dinkova-Kostova & Tew, 2020; Wang et al., 2023). In lung cancer, BACH1 promotes metastasis by upregulating antioxidant genes and enabling cancer cells to withstand high ROS levels (Li et al., 2021). Similarly, in breast cancer, BACH1 can suppress ferroptosis by modulating genes related to GSH synthesis and lipid metabolism (Kamble et al., 2021). Our study corroborates these findings by showing that BACH1 directly regulates GCLM, a rate-limiting enzyme in GSH biosynthesis, to promote ferroptosis resistance in ESCC. However, unlike previous studies that primarily focused on BACH1 as a standalone transcription factor, we reveal its functional cooperation with BAF53A, highlighting a broader regulatory network involving chromatin remodeling and transcriptional control. Interestingly, the cooperation between chromatin remodelers and transcription factors is not unique to BAF53A and BACH1. In other cancers, components of the SWI/SNF complex interact with lineage-specific transcription factors to regulate gene expression (Mittal & Roberts, 2020; Xiao et al., 2022). For instance, ARID1A, another SWI/SNF component, cooperates with transcription factors to regulate genes involved in tumor suppression and metabolism (Maxwell et al., 2024; Qu et al., 2019). Our findings position BAF53A and BACH1 within this broader context, emphasizing the significance of chromatin remodeling-transcription factor interactions in cancer biology.

GCLM, the modifier subunit of glutamate-cysteine ligase, is indispensable for GSH biosynthesis (Lu, 2013). Elevated GCLM expression has been noted in pancreatic, colorectal, and renal cancers, where it contributes to chemoresistance and ferroptosis evasion by enhancing antioxidant capacity (Lv et al., 2023; Wang et al., 2022b; Xie et al., 2023). For instance, in pancreatic cancer, GCLM upregulation confers resistance to gemcitabine (Patibandla et al., 2024), while in renal cancer, it is associated with ferroptosis suppression through enhanced detoxification of lipid peroxides (Gong et al., 2023). This study uncovers that BAF53A and BACH1 transcriptionally regulate GCLM in ESCC, forming a crucial axis that sustains GSH metabolism and prevents ferroptosis. These findings align with prior research but add a new dimension by identifying upstream chromatin and transcriptional regulators of GCLM, thereby expanding the understanding of its role in cancer biology.

In conclusion, this article uncovers an interaction between BAF53A and BACH1 that integrates chromatin remodeling and transcriptional regulation to drive GSH metabolism and ferroptosis resistance in ESCC. By elucidating the BAF53A-BACH1-GCLM axis, we provide new insights into the metabolic and epigenetic adaptations that underpin ESCC progression. Targeting this axis may represent a promising therapeutic option to exploit ferroptosis-based vulnerabilities in ESCC and other cancers. Future studies are warranted to disclose the broader functional roles of this interaction and translate these findings into clinical applications.

Supplemental Information

Supplemental Information 1 The uncropped western blots.

Supplemental Information 2 Raw data.

Supplemental Information 3 Original western blots.

Supplemental Information 4 Original figures.

Supplemental Information 5 MIQE checklist.

Additional Information and Declarations

Competing Interests

The authors declare that they have no competing interests.

Author Contributions

Weijuan Jiang conceived and designed the experiments, analyzed the data, prepared figures and/or tables, and approved the final draft.

Jie Zhang performed the experiments, authored or reviewed drafts of the article, and approved the final draft.

Canjuan Chen conceived and designed the experiments, prepared figures and/or tables, authored or reviewed drafts of the article, and approved the final draft.

Jiangwei Shi performed the experiments, prepared figures and/or tables, and approved the final draft.

Lihua Fan conceived and designed the experiments, performed the experiments, analyzed the data, prepared figures and/or tables, authored or reviewed drafts of the article, and approved the final draft.

Data Availability

The following information was supplied regarding data availability:

The raw data is available in the Supplemental Files and at figshare: Fan, Lihua (2025). Original Figures.zip. figshare. Figure. https://doi.org/10.6084/m9.figshare.29958014.v1.

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
