# Peer review of "The BAF53A-BACH1-GCLM axis regulates glutathione metabolism and enhances ferroptosis resistance in esophageal squamous cell carcinoma"

_PeerJ, doi:10.7717/peerj.20156_

## Round 0.1 · original submission · Major Revisions

It is my opinion as the Academic Editor for your article - The BAF53A-BACH1-GCLM Axis Drives GSH Metabolism and Ferroptosis Resistance in Esophageal Squamous Cell Carcinoma - that it requires some revision.

**Language Note:** The review process has identified that the English language must be improved. PeerJ can provide language editing services - please contact us at [email protected] for pricing (be sure to provide your manuscript number and title). Alternatively, you should make your own arrangements to improve the language quality and provide details in your response letter. – PeerJ Staff

Reviewer 1 ·

Basic reporting

1: The article uses some terms and abbreviations without providing their full forms. This may pose a challenge for non-experts or those unfamiliar with the field to understand. It is recommended to provide the full form of all terms upon their first appearance.

2: The authors claimed the use of "xenograft models" in Abstract. However, the description of this model is not mentioned in any place of the other text. Although tumor tissue examinations are presented in Fig 1 G-J, the corresponding description is missing.

Experimental design

1: Before performing BAF54A knockdown experiments in the cancer cell lines, did the authors examine the basal expression level of BAF54A in cancer cells relative to non-cancer cells (e.g. normal esophageal epithelial cells?). While the bioinformatics insights can provide supportive information, the precise gene expression pattern requires examination in tissues or cells.

2: In Fig 4, the authors suggest that "Given that BACH1 is a known transcriptional regulator of GSH-related genes and a modulator of ferroptosis [21, 25, 26], we hypothesized that BAF53A interacts with BACH1 to regulate GCLM transcription. BACH1 knockdown visibly reduced GCLM expression (Fig. 4E-F)" and "Co-IP assays confirmed the physical interaction between BAF53A and BACH1, and nuclear colocalization was validated through immunofluorescence (Fig. 4I-J) " However, there is not experimental evidence supporting the binding of BAF53A or BACH1 to the promoter region of GCLM.

Validity of the findings

0At the opening of Result 4, the authors claimed that " Among the GSH synthesis-related genes, GCLM showed the most pronounced downregulation in BAF53A-silenced cells (Fig. 4A-B)". This description does not precisely match the images as only GCLM was tested in Fig 4A-B.

Additional comments

In the manuscript, the authors report a novel BAF53A-BACH1-GCLM axis that drives GSH metabolism and ferroptosis resistance in esophageal squamous cell carcinoma. They claim that BAF53A, upregulated in ESCC, interacts with BACH1 and binds to the promoter of the GSH metabolism-related gene GCLM. Overall, the design and results of the research are credible and scientific. In the manuscript, the authors report a novel BAF53A-BACH1-GCLM axis that drives GSH metabolism and ferroptosis resistance in esophageal squamous cell carcinoma. They claim that BAF53A, upregulated in ESCC, interacts with BACH1 and binds to the promoter of the GSH metabolism-related gene GCLM. Overall, the design and results of the research are credible and scientific. However, several concerns should be addressed to enhance the comprehensiveness and scientific value of the paper

Annotated reviews are not available for download in order to protect the identity of reviewers who chose to remain anonymous.

Reviewer 2 ·

Basic reporting

The manuscript entitled "The BAF53A-BACH1-GCLM Axis Drives GSH Metabolism and Ferroptosis Resistance in Esophageal Squamous Cell Carcinoma" is interesting to the scientific community; however, mostly because of not clear writing the manuscript is confusing in places and the message is lost. In particular, most experimental images, most are related to animal experiments, are not mentioned in the text throughout the manuscript. Additionally, the descriptions for some results are oversimplified.

1: The text is overall understandable; however, the manuscript would be benefited from a professional language editing to improve the readability.

2: I recommend the authors to provide the catalog numbers for cell lines and key commercial regimens in the text.

3: In the "Data Analysis Methods" section, the authors claimed that "Independent sample t-tests were adopted for pairwise comparisons. One-way ANOVA was employed for multi-group comparisons". However, these comparisons seem inappropriate for data analysis in several contexts like 1D-1J and so forth. The appropriate method might be two-way ANOVA, which I could see from the figure legend. If so, this issue should be clarified in the method section.

4: Obvious mismatch between Images and texts: Fig 1 includes panels A-J; however, panels from 1G-1J are not described in the text. The detailed description for the images should be provided.

5: The description for results is oversimplified. For instance, at the opening of Result 2, the authors claimed that "GSH-metabolic genes in BAF53A-silenced ESCC cells were downregulated". Although readers can tell from Fig 2A that many genes were involved, these specific genes should be mentioned in the main text.

6: At the opening of Result 4, the authors claimed that " Among the GSH synthesis-related genes, GCLM showed the most pronounced downregulation in BAF53A-silenced cells (Fig. 4A-B)". This description does not precisely match the images as only GCLM was tested in Fig 4A-B.

7: Fig 3: Still, the description for panel 3H, which should be the positive staining of 4-HNE, was not described in the text. What tissues are these?

Experimental design

Please describe the experimental findings in a more specific and detailed way. For instance, the statement " BACH1 knockdown in ESCC cells recapitulated the phenotypes observed with BAF53A silencing, including diminished cell viability, reduced colony formation, increased ROS levels, and decreased GSH/GSSG and NADP+/NADPH ratios (Fig. 5A-E) " is too simplified to be accepted as an appropriate description in scientific documents.

Validity of the findings

no comment

Additional comments

I recommend the authors to address the limitations and future perspectives of this study.

Annotated reviews are not available for download in order to protect the identity of reviewers who chose to remain anonymous.

---

## Round 0.2 · accepted · Accept

The revision of this manuscript solves my problem and can be published.

Reviewer 1 ·

Basic reporting

no comment

Experimental design

no comment

Validity of the findings

no comment

Additional comments

no comment

Reviewer 2 ·

Basic reporting

The current form is acceptable.

Experimental design

The current form is acceptable.

Validity of the findings

The current form is acceptable.